# C-Phycocyanin Attenuates Noise-Induced Cochlear Synaptopathy via the Inhibition of Oxidative Stress and Intercellular Adhesion Molecule-1 in the Cochlea

**DOI:** 10.3390/ijms25105154

**Published:** 2024-05-09

**Authors:** Yi-Chun Lin, Cheng-Ping Shih, Yuan-Yung Lin, Hung-Che Lin, Chao-Yin Kuo, Hang-Kang Chen, Hsin-Chien Chen, Chih-Hung Wang

**Affiliations:** 1Department of Otolaryngology-Head and Neck Surgery, Tri-Service General Hospital, National Defense Medical Center, Taipei 11490, Taiwan; lyc_1023@yahoo.com.tw (Y.-C.L.); yking1109@gmail.com (Y.-Y.L.); lhj50702@gmail.com (H.-C.L.); chefsketchup@hotmail.com (C.-Y.K.); hwalongchen@yahoo.com.tw (H.-K.C.); acolufreia@yahoo.com.tw (H.-C.C.); chw@ms3.hinet.net (C.-H.W.); 2Division of Otolaryngology, Taipei Veterans General Hospital Taoyuan Branch, Taoyuan 33052, Taiwan

**Keywords:** C-phycocyanin, cochlear synaptopathy, inner hair cell, noise, oxidative stress

## Abstract

The synapses between inner hair cells (IHCs) and spiral ganglion neurons (SGNs) are the most vulnerable structures in the noise-exposed cochlea. Cochlear synaptopathy results from the disruption of these synapses following noise exposure and is considered the main cause of poor speech understanding in noisy environments, even when audiogram results are normal. Cochlear synaptopathy leads to the degeneration of SGNs if damaged IHC-SGN synapses are not promptly recovered. Oxidative stress plays a central role in the pathogenesis of cochlear synaptopathy. C-Phycocyanin (C-PC) has antioxidant and anti-inflammatory activities and is widely utilized in the food and drug industry. However, the effect of the C-PC on noise-induced cochlear damage is unknown. We first investigated the therapeutic effect of C-PC on noise-induced cochlear synaptopathy. In vitro experiments revealed that C-PC reduced the H_2_O_2_-induced generation of reactive oxygen species in HEI-OC1 auditory cells. H_2_O_2_-induced cytotoxicity in HEI-OC1 cells was reduced with C-PC treatment. After white noise exposure for 3 h at a sound pressure of 118 dB, the guinea pigs intratympanically administered 5 μg/mL C-PC exhibited greater wave I amplitudes in the auditory brainstem response, more IHC synaptic ribbons and more IHC-SGN synapses according to microscopic analysis than the saline-treated guinea pigs. Furthermore, the group treated with C-PC had less intense 4-hydroxynonenal and intercellular adhesion molecule-1 staining in the cochlea compared with the saline group. Our results suggest that C-PC improves cochlear synaptopathy by inhibiting noise-induced oxidative stress and the inflammatory response in the cochlea.

## 1. Introduction

Noise exposure is a common cause of acquired sensorineural hearing loss, and approximately 5% of the global population suffers from noise-induced hearing loss (NIHL) [1]. Noise exposure can cause damage to the cochlea and central auditory system [2]. Damage to the cochlea involves several regions, including the organ of Corti, spiral ganglion neurons (SGNs), and the spiral ligament. NIHL results in a temporary threshold shift and/or permanent threshold shift (PTS) in hearing. A high level of noise often leads to a prominent PTS and multiple lesions, including the death of cochlear hair cells, SGNs, and fibrocytes of the lateral wall [3]. The synapses between inner hair cells (IHCs) and type I SGNs are the structures most vulnerable to noise exposure and aging [4,5]. Although hearing thresholds completely recover and no hair cells are lost following exposure to low-to-moderate noise, cochlear synaptopathy manifesting as synaptic disruption between IHCs and SGNs may occur. Cochlear synaptopathy is recognized as the main pathogenesis of functional deficits in hearing in people with noise-induced hidden hearing loss [6]. These patients have normal audiogram results but suffer from difficulty understanding speech in noisy situations and other auditory symptoms. Animal studies have revealed that improving synaptic loss between IHCs and SGNs is crucial for the management of noise-induced hidden hearing loss [7,8]. There are several mechanisms contributing to the pathogenesis of NIHL. The formation of reactive oxygen species (ROS), which occurs during and after noise exposure, is regarded as a central mechanism leading to NIHL [3]. Previous studies have demonstrated that the administration of antioxidant agents has a protective effect on NIHL [9].

C-Phycocyanin (C-PC), which is extracted from Spirulina, has several biological functions, including antioxidant, anti-inflammatory and neuroprotective effects [10,11]. It is currently utilized as a dietary supplement, cosmetic agent, and pharmaceutical. C-PC exhibits antioxidant effects by scavenging ROS, inhibiting lipid peroxidation, and activating antioxidant enzymes [10,11]. According to previous animal studies, C-PC prevents cardiac damage caused by acute myocardial infarction, chemical toxicity, alcohol-induced hepatorenal injury, and diabetic nephropathy in animals by inhibiting oxidative stress and inflammation [12,13,14,15,16,17]. Furthermore, C-PC has therapeutic effects on colitis, acute lung injury, and the demyelination of the ischemic brain [18,19,20]. Kim et al. demonstrated the protective effect of C-PC on cisplatin-induced cytotoxicity in HEI-OC1 auditory cells [21]. Pretreatment with C-PC inhibited the expression of the proapoptotic protein Bax and enhanced the expression of Bcl-2 in the cells. C-PC protects cells against cisplatin toxicity by reducing intracellular ROS. Hwang et al. reported that oral administration of C-PC can reduce the expression of inflammation-associated genes in the cochlea and inferior colliculus that exhibit increased expression in a mouse model of salicylate-induced tinnitus [22]. This finding suggested that C-PC can exert anti-inflammatory effects on the cochlea. However, it is not known whether C-PC can mitigate noise-induced ROS generation and damage in the cochlea. In this study, we first demonstrated that the intratympanic administration of C-PC attenuates oxidative stress in the cochlea and cochlear synaptopathy in noise-exposed guinea pigs.

## 2. Results

### 2.1. C-PC Alleviates H_2_O_2_-Induced Cytotoxicity and Oxidative Stress in Auditory Cells

HEI-OC1 cells were treated with various concentrations of C-PC for 48 h to evaluate the impact of C-PC on cell viability (Figure 1A). The administration of C-PC at concentrations ranging from 1 to 5 μg/mL did not lead to cytotoxicity in HEI-OC1 cells. The protective effect of C-PC on H_2_O_2_-induced cytotoxicity was investigated (Figure 1B). The cell viability in the groups treated with H_2_O_2_ alone and those cotreated with H_2_O_2_ and C-PC was compared. A significant difference was noted between the groups treated with H_2_O_2_ alone and those treated with H_2_O_2_ and 5 μg/mL C-PC (49.7% ± 2.04% vs. 73.3% ± 4.02%, *p* = 0.0003). The administration of 5 μg/mL C-PC rescued the viability of HEI-OC1 cells after H_2_O_2_ treatment. These findings suggested that C-PC protected HEI-OC1 cells from H_2_O_2_-induced cytotoxicity. C-PC (5 μg/mL) was used for the following experiments. Oxidative stress plays a crucial role in the pathogenesis of NIHL [3]. Subsequently, the antioxidant effect of C-PC on H_2_O_2_-induced ROS in HEI-OC1 cells was investigated (Figure 2). ROS levels were increased in the cells following H_2_O_2_ treatment. C-PC significantly reduced ROS in H_2_O_2_-treated cells (*p* = 0.002). Our previous study showed that an upregulated expression of cochlear NADPH oxidase 4 (NOX4) contributes to ROS generation after noise exposure [23]. Furthermore, other studies have reported that C-PC inhibits NOX4 expression to reduce NADPH-induced superoxide formation [17,24]. Therefore, NOX4 expression in HEI-OC1 cells after H_2_O_2_ treatment and after H_2_O_2_ and C-PC treatment was compared in the present study (Figure 3). NOX4 mRNA levels in cells after H_2_O_2_ treatment increased 1.66-fold compared with that in untreated cells (control) (Figure 3A). NOX4 mRNA levels in cells after treatment with H_2_O_2_ and C-PC increased 1.09-fold compared to the control group. There was a significant difference in the mRNA expression of NOX4 between cells after H_2_O_2_ treatment and after treatment with H_2_O_2_ and C-PC (*p* = 0.02) (Figure 3A). Immunostaining for NOX4 in H_2_O_2_-treated cells was more intense than that in control cells (Figure 3B). Compared to that in the group after H_2_O_2_ treatment, the immunostaining intensity for NOX4 in the group after H_2_O_2_ and C-PC treatment was obviously decreased. C-PC reduced the H_2_O_2_-induced expression of the NOX4 gene and protein in auditory cells. The activities of intracellular antioxidant enzymes, including glutathione peroxidase (GPx), superoxide dismutase (SOD), and catalase (CAT), were compared in HEI-OC1 cells after H_2_O_2_ and C-PC treatment (Figure 4). There was no significant difference in the activity of these antioxidant enzymes between the groups treated with H_2_O_2_ alone and those treated with H_2_O_2_ and C-PC. C-PC can react with oxygen free radicals and is an efficient scavenger of ROS [10]. Taken together, these findings demonstrated that C-PC has the ability to scavenge ROS to improve H_2_O_2_-induced cytotoxicity and oxidative stress in auditory cells.

### 2.2. Intratympanic Administration of C-PC Rescued Noise-Induced Cochlear Synaptopathy

The ABR wave I amplitude is utilized to indicate the function of synapses between IHCs and SGNs [4]. To investigate the therapeutic effect of C-PC on noise-induced cochlear synaptopathy, ABR wave I amplitudes greater than 16, 24, and 32 kHz in the treatment and control groups were measured 28 days after noise exposure (Figure 5). The amplitudes over 24 and 32 kHz in the C-PC5 group were significantly greater than those in the saline group (24 kHz, *p* = 0.02; 32 kHz, *p* = 0.02). There was no significant difference in response amplitude between the C-PC1 and saline groups. These findings suggested that the animals that received intratympanic administration of 5 μg/mL C-PC had less damage to the IHC-SGN synapses. Carboxyl-terminal binding protein 2 (CtBP2) was utilized to identify synaptic ribbons with IHCs, and the distribution of CtBP2 puncta in the three groups was compared using IHCs on Day 28 after noise exposure (Figure 6). The number of IHC synaptic ribbons in the three groups was as follows: 13 ± 0.74 in the basal turn of the saline group, 11 ± 0.9 in the second turn of the saline group, 15.3 ± 0.89 in the basal turn of the C-PC1 group, 12.7 ± 0.62 in the second turn of the C-PC1 group, 17.1 ± 0.5 in the basal turn of the C-PC5 group and 14.2 ± 0.76 in the second turn of the C-PC5 group. There was a greater number of synaptic ribbons in the C-PC5 group than in the saline group (basal turn, *p* = 0.0004; second turn, *p* = 0.015). The C-PC5 group exhibited a more significant preservation of IHC synaptic ribbons than the saline group. IHC-SGN synapses were identified as puncta with colocalization of staining for CtBP2 and glutamate receptor subunit A2 (GluA2). The distribution of IHC-SGN synapses was evaluated in the three groups on Day 28 after noise exposure (Figure 7). The number of synapses per IHC in the three groups was as follows: 7.9 ± 1.2 in the basal turn of the saline group, 8.4 ± 0.92 in the second turn of the saline group, 11.7 ± 1.06 in the basal turn of the C-PC1 group, 9.9 ± 0.86 in the second turn of the C-PC1 group, 13.6 ± 0.77 in the basal turn of the C-PC5 group and 11.7 ± 0.91 in the second turn of the C-PC5 group. There were more IHC-SGN synapses in the basal and second turns in the C-PC5 group than in the saline group (basal turn, *p* = 0.001; second turn, *p* = 0.02). The C-PC1 group had a greater number of IHC-SGN synapses in the basal turn than did the saline group (*p* = 0.03). These findings demonstrated that C-PC5 treatment preserved the IHC-SGN synapses well after noise exposure.

Subsequently, 4-hydroxynonenal (4-HNE), a marker of lipid peroxidation and oxidative stress, was utilized to determine ROS levels in the cochleae of the three groups 48 h after noise exposure (Figure 8A). The saline group exhibited intense staining in the spiral ligament and spiral limbus. Among the three groups, the C-PC5 group exhibited the weakest staining in the spiral ligament and spiral limbus, followed by the C-PC1 group. These findings suggested that C-PC treatment diminished noise-induced ROS generation in the cochlea. Cochlear inflammation contributes to NIHL, and intercellular adhesion molecule-1 (ICAM-1) is a marker for noise-induced cochlear inflammation [25,26]. After noise exposure, ICAM-1 was detected in the spiral ligament in the three groups (Figure 8B). The saline group had the most intense ICAM-1 staining, especially in the type II and IV fibrocytes of the spiral ligament. Weak staining was noted in the C-PC1 and C-PC5 groups. These findings suggested that C-PC rescued noise-induced cochlear synaptopathy by reducing oxidative stress and inflammation in the cochlea.

## 3. Discussion

ROS are immediately induced at an intense level when the cochlea is exposed to harmful noise. During noise exposure, overdriving of the mitochondria in the cochlea and excitotoxicity between IHCs and SGNs create large amounts of superoxide [3]. Higher levels of other ROS can be generated from the reaction of increased superoxide and other substances. Moreover, high-level noise exposure can reduce cochlear blood flow [3]. Ischemia and reperfusion injuries due to a reduction in cochlear blood flow result in increased ROS formation. An increase in ROS leads to DNA and protein damage and lipid peroxidation in cells [3]. Cell apoptosis and necrosis subsequently develop in the cochlea. On the other hand, there is a delayed formation of ROS in the cochlea days after noise exposure, which contributes to progressive cochlear lesions, including hair cell loss [23,27]. The excessive generation of ROS in the auditory system is also linked to other types of acquired sensorineural hearing loss, including cisplatin-induced ototoxicity and presbycusis [28]. C-PC is a potent free radical scavenger and can react with alkoxyl, hydroxyl, and peroxyl radicals [10]. An in vitro study reported the antioxidant effect of C-PC in HEI-OC1 cells [21]. Oral supplementation with *Spirulina platensis* extract, which contains C-PC, results in a reduction in lipid peroxidation and the activation of antioxidant enzymes in the cochleae of senescence-accelerated prone-8 mice [29]. The present study demonstrated that C-PC can scavenge H_2_O_2_-induced intracellular ROS in auditory cells. Moreover, C-PC attenuated cochlear oxidative stress in a guinea pig model of noise trauma. Our results support the use of C-PC as an antioxidant for the management of inner-ear diseases. Inflammation plays a role in the pathogenesis of NIHL [25,26,30]. Some anti-inflammatory agents have protective effects on NIHL. In this study, noise-induced ICAM-1 expression in the spiral ligament was inhibited by C-PC. Therefore, C-PC could exert an anti-inflammatory effect on the cochlea in NIHL.

Glutamate excitotoxicity due to acoustic overstimulation contributes to the disruption of IHC synaptic ribbons, which occurs early after noise exposure [31]. Kurasawa et al. reported that ROS lead to the degeneration of IHC synaptic ribbons with a decreased expression of Piccolo 1—a major component of synaptic ribbons [32]. ROS have been demonstrated to be one of the causes of cochlear synaptopathy in acquired sensorineural hearing loss. In noise-induced hidden hearing loss, synaptic dysfunction between IHCs and SGNs, such as difficulty in word recognition in a noisy background, is recognized as the leading cause of hearing disability [33]. Furthermore, noise-induced cochlear synaptopathy can lead to tinnitus when accompanied by altered neural plasticity in the dorsal cochlear nucleus following noise trauma [34]. In age-related hearing loss, the degeneration of IHC-SGN synapses occurs earlier than hair cell loss [8]. Cochlear synaptopathy progresses to the neurodegeneration stage and manifests as the loss of spiral ganglion cells if synapses are not recovered [4]. Neurodegeneration is an important factor contributing to the perceptual handicap in sensorineural hearing loss [8]. The role of the C-PC in the protection of cochlear synapses from any damage has not been previously investigated. The present study provided new findings of C-PC activity in the noise-exposed cochlea, including the protection of IHC-SGN synapses. ABR wave I amplitudes correlate with the integrity of IHC-SGN synapses in animal models [4]. In the present study, the wave I amplitudes of the animals intratympanically administered 5 μg/mL C-PC were greater than those of the saline-treated animals 28 days after noise exposure. Furthermore, according to our microscopic analysis, the administration of 5 μg/mL C-PC diminished IHC-SGN synapse loss in the animals in which the synapse number was approximately 1.5-fold greater than that in the saline group. These results revealed the protective effect of C-PC on IHC-SGN synapses. Moreover, C-PC administration diminished noise-induced oxidative stress and ICAM-1 expression in the cochlea. These findings demonstrated that C-PC can attenuate noise-induced cochlear synaptopathy by scavenging ROS and inhibiting ICAM-1. Excessive ROS formation is one main cause of noise-induced damage to IHC synaptic ribbons [32]. Therefore, the antioxidant activity of C-PC plays a crucial role in the protection of IHC-SGN synapses. ICAM-1 facilitates inflammatory response in the noise-exposed cochlea [25,26]. Since cochlear ICAM-1 expression can be inhibited by C-PC, C-PC exerts anti-inflammatory activity contributing to the protective effect of C-PC on the synapses. However, the effect of C-PC on inflammation-associated signaling pathways was not examined in this study and there is a need for further investigation.

Previous studies and this study investigated the protective effect of C-PC and Spirulina on drug-induced ototoxicity, the cochleae of senescence-accelerated prone-8 mice, and noise-induced cochlear synaptopathy [21,22,29,35,36,37,38,39]. These findings support that C-PC can protect the cochlea against different damage. C-PC preserves mitochondrial function and exerts an anti-apoptotic effect on cisplatin-induced cytotoxicity by reducing intracellular ROS and suppressing apoptotic pathway [21]. In rats with cisplatin-induced ototoxicity, Spirulina reduces apoptosis in the organ of Corti and improves hearing function [35]. In kanamycin-treated rats, the cochlea in the group with Spirulina treatment has less hair cell damage, fewer macrophage cells, and less vascular dilation than that without treatment [36]. In the animal model of salicylate-induced tinnitus, Spirulina and C-PC regulate the expression of several genes, including antioxidant, inflammatory, and ion co-transport genes in the cochlea [22,37,38,39]. The treatments lead to a reduction in tinnitus severity. In this study, C-PC protects auditory cells against H_2_O_2_-induced cytotoxicity by scavenging ROS and can attenuate noise-induced cochlear synaptopathy in our animal model. To sum up, C-PC has otoprotective effect and a potential for clinical application.

NIHL involves cellular damage from the cochlea to the central auditory system. Noise exposure can cause neural cell loss in the cochlear nucleus, central nucleus of the inferior colliculus, medial geniculate body, and primary auditory cortex [2]. Noise stress enhances lipid peroxidation in various regions of the brain, including the midbrain and brainstem [40]. These findings suggest that oxidative stress in the central auditory system contributes to NIHL. C-PC can improve ischemic brain injury and neuroinflammatory diseases in animals by reducing the oxidative stress response and inflammation of neurons [20,41]. There was an increase in the expression of N-methyl D-aspartate receptor subunit 2B, tumor necrosis factor-α, and interleukin-1β in the inferior colliculus of mice following salicylate treatment [22]. Oral administration of C-PC or *Spirulina platensis* extract can reduce the severity of salicylate-induced tinnitus and the expression of N-methyl D-aspartate receptor and inflammatory genes. Moreover, dietary supplementation with *Spirulina platensis* extract results in improved hearing thresholds in senescence-accelerated mice by reducing oxidative stress damage and activating antioxidant enzymes in the brainstem and cochlea [29]. According to the abovementioned findings, oral administration of C-PC may be helpful for protecting the central auditory system from noise-induced damage. Further studies will be conducted to clarify the protective effect of C-PC on the central auditory system.

## 4. Materials and Methods

### 4.1. Cell Culture

The use of HEI-OC1 cells, a mouse auditory cell line [42], was supported by Dr Federico Kalinec (House Ear Institute, Los Angeles, CA, USA). HEI-OC1 cells were cultured in Dulbecco’s modified Eagle’s medium (Gibco, Grand Island, NY, USA) supplemented with 10% fetal bovine serum (Biological Industries, Kibbutz Beit-Haemek, Israel) without antibiotics at 33 °C and 10% CO_2_. H_2_O_2_-induced cell damage was induced by treatment with 30 μM H_2_O_2_ for 24 h in HEI-OC1 cells. After treatment with C-PC (Agilent, Santa Clara, CA, USA), the cells were pretreated with 1 or 5 μg/mL C-PC for 24 h and then cotreated with H_2_O_2_ and C-PC for 24 h. The cells were collected for analysis 24 h after H_2_O_2_ treatment.

### 4.2. Animal Study

The experimental protocol (protocol No.: IACUC-18-221) was approved by the Institutional Animal Care and Use Committee of the National Defense Medical Center, Taipei, Taiwan. Animal care complied with institutional guidelines and regulations. A total of 42 pigmented male guinea pigs, weighing 250–350 g, with a normal Preyer’s reflex, were separated into three groups, including two treatment groups and one control group: (i) one treatment group treated with 10 μL of C-PC at a concentration of 1 μg/mL (C-PC1) 24 h after noise exposure, (ii) another treatment group treated with 10 μL of C-PC at a concentration of 5 μg/mL (C-PC5) 24 h after noise exposure, and (iii) a control group treated with 0.9% saline (the saline group) 24 h after noise exposure.

### 4.3. Surgical Procedures

Guinea pigs were anesthetized with 10 mg/kg xylazine (Rompun; Bayer, Leverkusen, Germany) and 80 mg/kg ketamine (KARSOON MINE 100, Taipei, Taiwan) intramuscularly and kept warm with a heating pad. A fenestration (approximately 4 mm in diameter) was created in the tympanic bulla by drilling with diamond burrs, and the round window membrane was exposed. A gelatin sponge (Johnson & Johnson, New Brunswick, NJ, USA) was cut into small pieces 2 mm^3^ in size. One piece of gelatin sponge was soaked in 10 μL of 0.9% saline or C-PC at a concentration of 1 or 5 μg/mL. A piece of drug-soaked gelatin sponge was placed on the round window membrane. All procedures were performed with an operating microscope (F-170; Carl Zeiss, Jena, Germany). The surgical wound was sutured in layers.

### 4.4. Noise Exposure

Guinea pigs were anesthetized, placed in a soundproof booth with a loudspeaker (V12 HP, Tannoy, Coatbridge, UK) mounted above the center of the cage, and exposed to 118 dB sound pressure level white noise for 3 h. The noise level was measured using a sound level meter (Rion NL-52, Tokyo, Japan). The difference in the noise level within the booth near the center or edge of the cage was less than 1 dB.

### 4.5. Evaluation of Cell Viability

To determine cell viability, the cell proliferation reagent WST-1 (Merck KGaA, Darmstadt, Germany) was added to the cell suspension in each well and incubated for 4 h. The reaction was catalyzed by a mitochondrial reductase in active cells, and the amount of formazan dye could be quantified by measuring the absorbance at 450 nm using an ELISA microplate reader (Synergy H4 Hybrid Reader, Agilent Technologies Inc., Santa Clara, CA, USA) to calculate the optical density (OD) values (A450–A655 nm).

### 4.6. Measurement of ROS Levels

Cellular ROS levels were measured using the fluorescent dye DCFH-DA (D399, Thermo Fisher Scientific Inc., Waltham, MA, USA). Briefly, cells were seeded in 24-well plates and treated under the indicated conditions. After washing with phosphate-buffered saline (PBS), a medium containing 20 μM DCFH-DA was added to each well, and samples were incubated at 33 °C for 30 min. ROS levels were measured using a microplate ELISA reader (Synergy H4 Hybrid Reader, Agilent Technologies Inc., CA, USA) at an emission wavelength of 528 nm and an excitation wavelength of 485 nm. The relative ROS level is expressed as the change in fluorescence of the experimental groups compared with that of the controls.

### 4.7. Quantitative PCR

Total RNA was extracted from HEI-OC1 cells with a high-purity RNA isolation kit (Merck KGaA, Darmstadt, Germany) according to the manufacturer’s protocol. RNA was converted to cDNA using a QuantiNova Reverse Transcription Kit (QIAGEN, Hilden, Germany). Gene expression was measured with TaqMan gene expression assays (Thermo Fisher Scientific Inc., Waltham, MA, USA) for NOX4 (probe ID: Mm00479246_m1) and glyceraldehyde-3-phosphate dehydrogenase (GAPDH; probe ID: Mm99999915_g1) using a QuantiNova Probe RT–PCR Kit (QIAGEN, Hilden, Germany) and a QuantStudio 5 Real-Time PCR system (Thermo Fisher Scientific Inc., Waltham, MA, USA). The qPCR data are presented as the gene expression level relative to the level in the controls after normalization to the expression of GAPDH.

### 4.8. Immunofluorescence Staining of NOX4

The cells were fixed with freshly prepared 4% paraformaldehyde at 37 °C for 30 min and permeabilized with 0.1% Triton X-100 for 5 min in BlockPRO blocking buffer (Visual Protein Biotechnology, Taipei, Taiwan). The cells were incubated with anti-NOX4 (1:100; NB110-58849, Novus Biologicals, Centennial, CO, USA) for 1 h at room temperature (RT) and then incubated with secondary antibodies against NOX4 (Alexa Fluor 488 donkey anti-rabbit, 1:500; A21206, Thermo Fisher Scientific, Waltham, MA, USA) for 1 h. The cells were mounted in 4,6-diamidino-2-phenylindole (DAPI) Fluoromount-G^®^ mounting medium (SouthernBiotech, Birmingham, AL, USA). Cell images were captured with a confocal laser scanning microscope (Zeiss LSM 880, Carl Zeiss, Jena, Germany).

### 4.9. Measurement of the Activity of Intracellular Antioxidant Enzymes

After centrifugation, the supernatants from the cell lysates were collected for analysis. The activity of GPx was measured using a GPx colorimetric assay kit (Abcam, Cambridge, UK). Briefly, the sample was incubated with the reaction mixture for 15 min at RT. After adding cumene hydroperoxide solution to the sample, the absorbance of the sample was measured at 340 nm for 5 min on a microplate reader at RT. GPX activity was calculated from the standard curve. SOD activity was measured using a SOD colorimetric assay kit (Abcam, Cambridge, UK). The sample was mixed with enzyme working solutions and incubated for 20 min at 37 °C. The absorbance of the sample was measured at 450 nm, and SOD activity was measured. The activity of CAT was measured using a CAT assay kit (Cayman, Ann Arbor, MI, USA). After the assay buffer and methanol were added, the sample was incubated with hydrogen peroxide for 20 min at RT. Then, potassium hydroxide was added to stop the reaction. After the sample was incubated with CAT Purpald (Chromogen, Dingley Village, VIC, Australia) for 10 min at RT, the absorbance was measured at 54 nm. CAT activity was calculated from the standard curve.

### 4.10. Auditory Brainstem Response Recording

Animal auditory function was assessed by recording ABRs as described previously [43]. In brief, guinea pigs were anesthetized and kept warm with a heating pad in a sound-attenuating chamber. Subdermal needle electrodes were inserted at the vertex (positive), below the pinna of the ear (negative), and at the back (ground) of the guinea pigs. Specific stimuli (16, 24, and 32 kHz tone bursts) were generated using SigGen software (Tucker-Davis Technologies, Gainseville, FL, USA) and delivered to the external auditory canal. ABR wave I was identified at a 90 dB HL stimulus of tone bursts. The wave I amplitude was calculated as the difference between the peak of wave I and the following trough.

### 4.11. Cochlear Surface Preparations

Animals were transcardially perfused with 4% paraformaldehyde in 0.1 M PBS following flushing with prewarmed PBS. The cochleae of the guinea pigs were removed and immersed in 4% paraformaldehyde for 1 h at RT. The organ of Corti was carefully dissected and permeabilized with 5% horse serum in PBS supplemented with 0.3% Triton X-100 for 1 h. The samples were then incubated with anti-CtBP2 IgG1 (1:100; 612044, BD Biosciences, San Jose, CA, USA), anti-GluA2 IgG (1:1000; MAB397, Merck KGa, Darmstadt, Germany) and anti-myosin 7a (1:100; sc-74516, Santa Cruz Biotechnology Inc., Dallas, TX, USA) polyclonal antibodies with 3% horse serum in PBS supplemented with 0.3% Triton X-100 overnight at 37 °C. After three washes with PBS, the tissues were incubated with secondary antibodies against CtBP2 (Alexa Fluor™ 555 goat anti-mouse IgG1, 1:500, A21127, Thermo Fisher Scientific, Waltham, MA, USA), GluA2 (Alexa Fluor™ 488 goat anti-mouse IgG2a, 1:1000, A21131, Thermo Fisher Scientific, Waltham, MA, USA), and myosin 7a (Alexa Fluor™ 647 donkey anti-rabbit IgG (H+L), 1:500, A31573, Thermo Fisher Scientific, Waltham, MA, USA) for 1 h at 37 °C. The samples were mounted with DAPI Fluoromount-G^®^ mounting medium (SouthernBiotech, Birmingham, AL, USA) and covered with a coverslip for analysis. Images were obtained using an LSM 880 Zeiss confocal microscope.

### 4.12. Cryosections of Cochleae

Following the gentle perfusion of 4% paraformaldehyde in PBS into the cochleae through the opened oval window and a small hole in the apex, the cochleae were postfixed for 2 h at RT. After fixation, the cochleae were washed with 0.1 M PBS thrice and then transferred to 10% ethylene diamine tetraacetic acid (EDTA), pH 7.3, at 4 °C with rotation for decalcification. The EDTA solution was changed daily until decalcification was complete. Decalcified cochleae were washed thrice in PBS and then immersed in 10% sucrose with rotation at RT for 30 min, followed by 15%, 20%, and 25% sucrose for 30 min, after which the sucrose was replaced with 30% sucrose at 4 °C with rotation overnight. The next day, the cochleae were transferred to OCT and placed into a vacuum desiccator under a gentle vacuum overnight to ensure the removal of all the air bubbles. A stereomicroscope was used to orient the cochleae, and the samples were transferred to a freezing slurry of solid CO_2_ and 100% ethanol for 7–10 s until the tissue was immobilized and solidified. Using a cryostat microtome, 14-micron mid-modiolar sections of frozen cochleae were mounted on glass slides.

### 4.13. Immunohistochemistry

Immunohistochemical staining was performed using the Mouse/Rabbit PolyDetector HRP/DAB Detection System (Bio SB Inc., Santa Barbara, CA, USA). The frozen slides with cryostat sections were allowed to warm at RT for 30 min, and the slides were rehydrated in wash buffer (1× PBS/0.1% Tween-20) for 15 min. Endogenous peroxidase activity was quenched with a PolyDetector Peroxidase Blocker for 15 min. After washing with wash buffer, nonspecific antibody binding was blocked by preincubating the slides with blocking buffer (1× PBS/3% horse serum/0.3% Triton™ X-100) for 1 h at RT. After blotting, the slides were incubated with mouse anti-4-HNE (1:100, ab46545, Abcam, Cambridge, UK) and anti-ICAM-1 (1:100, 14-0542-82, Thermo Fisher Scientific, Waltham, MA, USA) primary antibodies in antibody dilution buffer (Dako Co., Carpinteria, CA, USA) in a humidified chamber for 2 h at RT. After being washed in wash buffer thrice, the slides were incubated with the secondary antibody PolyDetector Label HRP for 30 min at RT. The slides were then immersed in PolyDetector DAB substrate-chromogen solution for 10 min. The slides were rinsed in distilled water, counterstained with hematoxylin (Muto Pure Chemicals Co., Ltd., Tokyo, Japan), dehydrated through a graded series of alcohol (50%, 70%, 80%, 90%, and 100%), cleared in xylene, and mounted in Permount (Fisher Scientific, Pittsburgh, PA, USA). The slides were examined using an Olympus BX50 microscope (Olympus Corporation, Hachioji-shi, Tokyo, Japan).

### 4.14. Statistical Analysis

The obtained data were analyzed statistically using Student’s *t*-test for comparisons between the two groups. Multiple groups were compared using one-way ANOVA followed by Scheffe’s multiple-comparisons test. The results are expressed as the mean ± standard error of the mean (SEM). Differences were considered significant at *p* < 0.05.

## 5. Conclusions

We confirmed the antioxidant effect of C-PC on auditory cells in vitro and in vivo. In our animal model of noise trauma, the intratympanic administration of C-PC improved the recovery of IHC-SGN synapses from noise-induced cochlear synaptopathy by attenuating oxidative stress and ICAM-1 expression. These findings suggest that C-PC could have clinical applications in the treatment of noise-induced cochlear synaptopathy. Moreover, oral supplementation with C-PC may be beneficial for preventing NIHL.

## Figures and Tables

**Figure 1 ijms-25-05154-f001:**
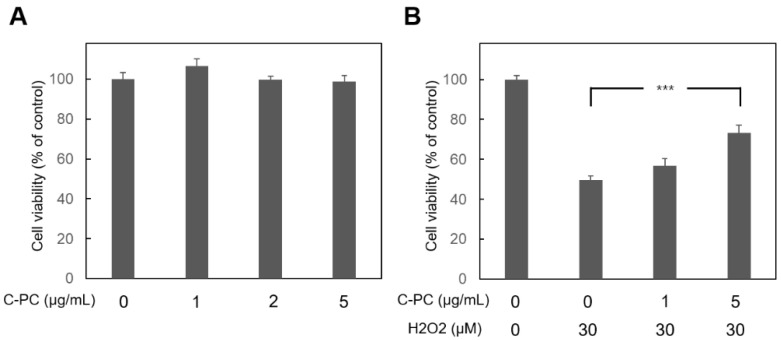
Effect of C-PC on the viability of HEI-OC1 cells without and with H_2_O_2_ treatment. Cell viability was evaluated using the WST-1 assay. (**A**) The cells were incubated with 0, 1, 2, or 5 μg/mL C-PC for 48 h. n = 6 for each group. There was no cytotoxicity after treatment with 5 μg/mL C-PC. (**B**) The cells were treated with C-PC for 24 h and then cotreated with H_2_O_2_ and C-PC for 24 h. The results are expressed as the mean ± SEM, with n = 8 for each group. *** *p* < 0.005.

**Figure 2 ijms-25-05154-f002:**
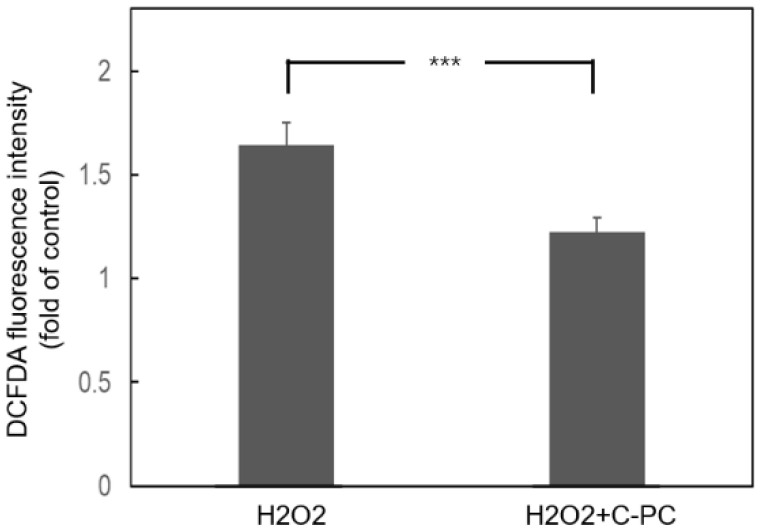
H_2_O_2_-induced ROS in HEI-OC1 cells were reduced by C-PC. ROS generation was measured using the fluorescent probe DCFDA. The results are expressed as the mean ± SEM, with n = 12 for each group. *** *p* < 0.005.

**Figure 3 ijms-25-05154-f003:**
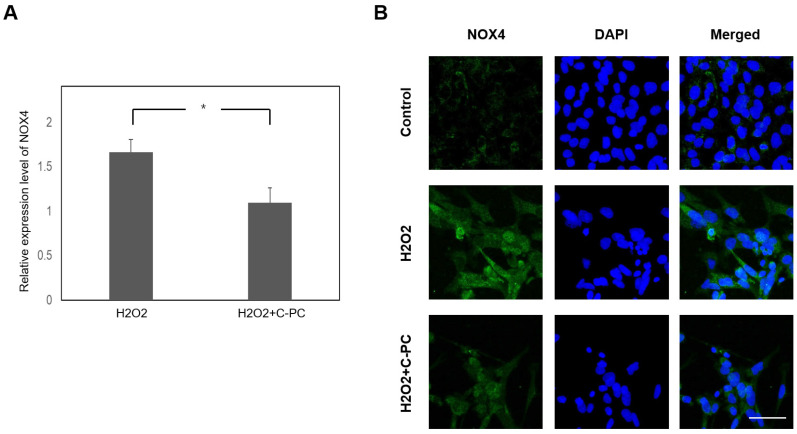
H_2_O_2_-induced NOX4 expression in HEI-OC1 cells was inhibited by C-PC. (**A**) NOX4 mRNA expression was measured using quantitative PCR. The cells were collected 8 h after H_2_O_2_ treatment. The results are expressed as the fold change relative to the control (the mean ± SEM), with n = 8 for each group. * *p* < 0.05. (**B**) Representative image of NOX4 immunostaining (green) in HEI-OC1 cells. Blue indicates DAPI. The cells were collected 24 h after H_2_O_2_ treatment. n = 6 for each group. Scale bar: 50 μm.

**Figure 4 ijms-25-05154-f004:**
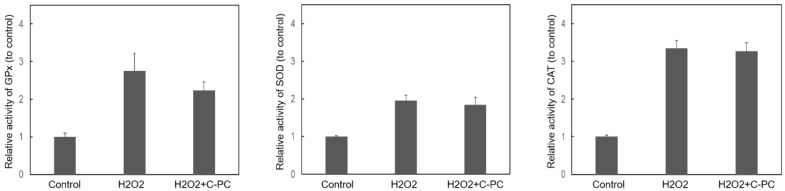
The activity of antioxidant enzymes, including glutathione peroxidase, superoxide dismutase and catalase, in HEI-OC1 cells was measured after H_2_O_2_ and C-PC treatment. The results are expressed as the mean ± SEM, with n = 6–8 for each group. GPx, glutathione peroxidase; SOD, superoxide dismutase; CAT, catalase.

**Figure 5 ijms-25-05154-f005:**
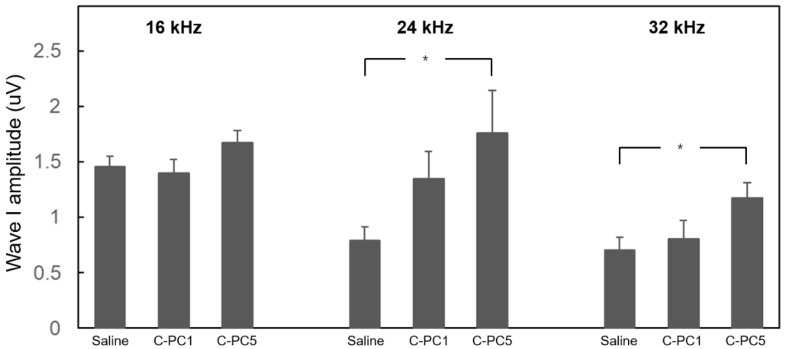
Comparison of ABR wave I amplitudes among the three groups 28 days after noise exposure. The wave I amplitudes were measured at 90 dB HL stimuli of 16, 24, and 32 kHz tone bursts. The results are expressed as the mean ± SEM, with n = 8 for each group. * *p* < 0.05.

**Figure 6 ijms-25-05154-f006:**
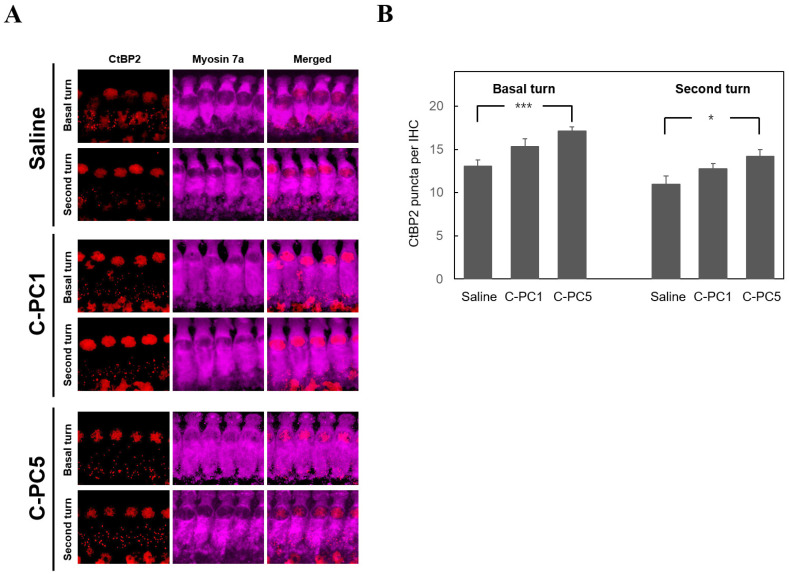
More synaptic ribbons were present in the IHCs of the groups receiving C-PC treatment. (**A**) Representative images of IHCs in the basal and second turns of the cochlea in the three groups on Day 28 after noise exposure. Immunofluorescence staining showing CtBP2 (red) and myosin 7a (violet). Scale bar: 100 μm. (**B**) Histograms summarizing the number of CtBP2 puncta per IHC at different turns. The results are expressed as the mean ± SEM, with n = 10 for each group. *** *p* < 0.005, * *p* < 0.05. IHC, inner hair cell.

**Figure 7 ijms-25-05154-f007:**
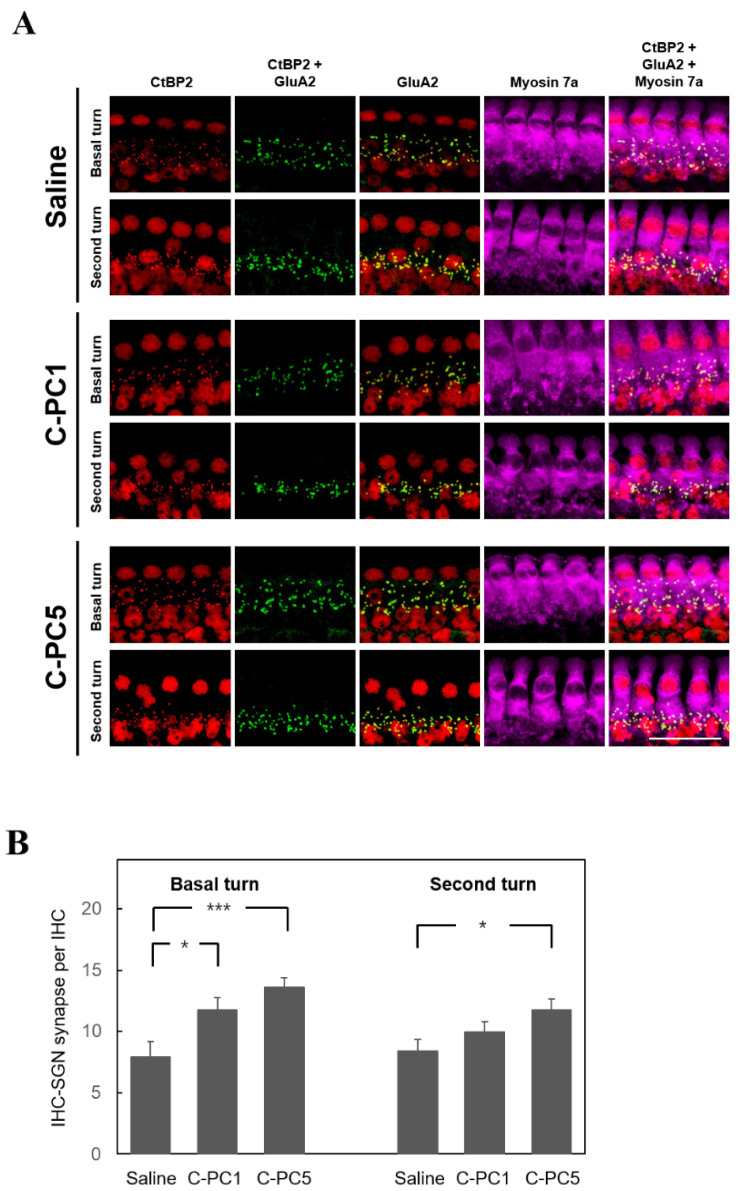
C-PC treatment attenuated the loss of synapses between IHCs and SGNs resulting from noise trauma. (**A**) Representative images of IHCs in the basal and second turns of the cochlea in the three groups on Day 28 after noise exposure. Immunofluorescence staining showing CtBP2 (red), GluA2 (green), and myosin 7a (violet). Yellow indicates colocalization of CtBP2 and GluA2. Scale bar: 100 μm. (**B**) Histograms summarizing the number of IHC-SGN synapses at different turns. The results are expressed as the mean ± SEM, with n = 10 for each group. *** *p* < 0.005, * *p* < 0.05. IHC, inner hair cell; SGN, spiral ganglion neuron.

**Figure 8 ijms-25-05154-f008:**
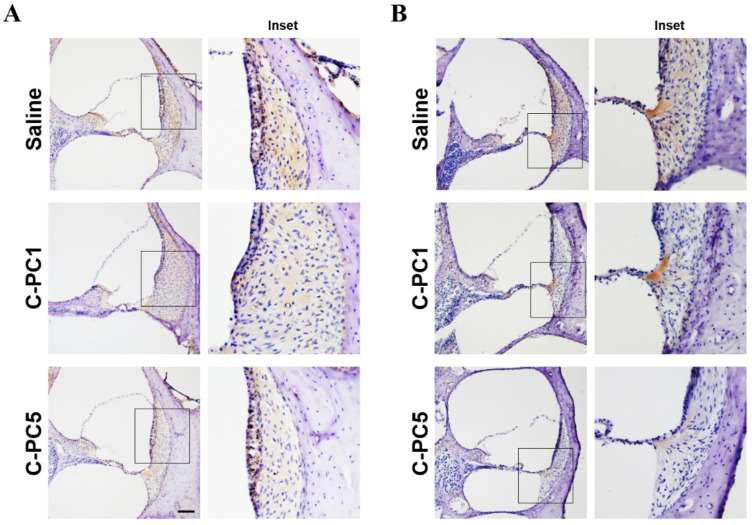
C-PC treatment attenuated noise-induced ROS generation and ICAM-1 expression in the cochlea. Representative immunohistochemical staining (brown color) for (**A**) 4-HNE and (**B**) ICAM-1 in the cochleae 48 h after noise exposure in the three groups. The basal turns of mid-modiolar sections of the cochleae are shown. The inset figure shows the region of the spiral ligament. n = 4 for each group. Scale bar: 100 μm.

## Data Availability

All relevant data are included within the manuscript. The raw data are available on request from the corresponding author.

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
