# Peer review of "C-Phycocyanin Attenuates Noise-Induced Cochlear Synaptopathy via the Inhibition of Oxidative Stress and Intercellular Adhesion Molecule-1 in the Cochlea"

_ijms, 2024, doi:10.3390/ijms25105154_

Round 1

Reviewer 1 Report

Comments and Suggestions for Authors

Study on C-Phycocyanin Reducing Noise-Induced Cochlear Synaptopathy via Inhibition of Oxidative Stress in the Cochlea. This paper is very impressive about detecting the operating problem and his work on the section. It is recommended to review some elements of the study.

A stronger literature contribution. Supporting the study with more references. The fact that these studies are current studies may emphasize the importance of the study.

It may be recommended to use colorful and lighter graphics.

The article's organization needs revision. The journal's Author guide can help with this. Using the titles Literature - Method - Findings - Results and Evaluation in order may comfort the reader. Journal instructions for the use of abbreviations can be followed. Figures 3-6-7 can be made more understandable. Ethics committee support can be added to the method section.

Author Response

ijms-2979996 titled " C-Phycocyanin attenuates noise-induced cochlear synaptopathy via the inhibition of oxidative stress and intercellular adhesion molecule-1 in the cochlea "

Point-by-point responses to reviewer comments:

  1. Study on C-Phycocyanin Reducing Noise-Induced Cochlear Synaptopathy via Inhibition of Oxidative Stress in the Cochlea. This paper is very impressive about detecting the operating problem and his work on the section. It is recommended to review some elements of the study.

A stronger literature contribution. Supporting the study with more references. The fact that these studies are current studies may emphasize the importance of the study.

Reply: Thanks for your comment and suggestion. More discussion and references were added in the revised manuscript.

In page 9 line 258 – 272 of the revised manuscript (changes are highlighted in red):

Previous studies and this study investigate the protective effect of C-PC and Spirulina on drug-induced ototoxicity, the cochleae of senescence-accelerated prone-8 mice and noise-induced cochlear synaptopathy [21,22,30,36-40]. These findings support that C-PC can protect the cochlea against different damage. C-PC preserves mitochondrial function and exerts anti-apoptotic effect on cisplatin-induced cytotoxicity by reducing intracellular ROS and suppressing apoptotic pathway [21]. In rats with cisplatin-induced ototoxicity, Spirulina reduces apoptosis in the organ of Corti and improves hearing function [36]. In kanamycin-treated rats, the cochlea in the group with Spirulina treatment has less hair cell damage, fewer macrophage cells and less vascular dilation than that without treatment [37]. In the animal model of salicylate-induced tinnitus, Spirulina and C-PC regulate the expression of several genes, including antioxidant, inflammatory and ion co-transport genes in the cochlea [22,38-40]. The treatments lead to a reduction in tinnitus severity. In this study, C-PC protects auditory cells against H2O2-induced cytotoxicity by scavenging ROS and can attenuate noise-induced cochlear synaptopathy in our animal model. To sum up, C-PC has otoprotective effect and a potential for clinical application.

In page 14 line 530 – page 15 line 539 of the revised manuscript (changes are highlighted in red):

  1. Tahir E, Buyuklu AF, Ocal FCA, Gurgen SG, Sarsmaz Protective effect of Spirulina on cisplatin-induced ototoxicity: a functional and histopathological study. B-ENT. 2022;18(1):34-43.
  2. Harinto PE, Ruspita DA, Marliyawati D, Widodo P, Naftali Z. Effect of Spirulina on cochlea histopathological changes in Wistar rats induced by kMal J Med Health Sci. 2023:19(2):69-75.
  3. Hwang JH, Chang NC, Chen JC, Chan YC. Expression of antioxidant genes in the mouse cochlea and brain in salicylate-induced tinnitus and effect of treatment with Spirulina platensis water e Audiol Neurootol. 2015;20(5):322-9.
  4. Chan YC, Wang MF, Hwang JH. Effects of Spirulina on GABA receptor gene expression in salicylate-induced tinnitus. Int Tinnitus J. 2018;22(1):84-8.
  5. Hwang JH, Chan YC. Expressions of ion co-transporter genes in salicylate- induced tinnitus and treatment effects of spirulina. BMC Neurol. 2016;16(1):159.

  1. It may be recommended to use colorful and lighter graphics.

Reply: Thanks for your comment. Figure 3, 6 and 7 were revised in the revised manuscript.

  1. The article's organization needs revision. The journal's Author guide can help with this. Using the titles Literature - Method - Findings - Results and Evaluation in order may comfort the reader. Journal instructions for the use of abbreviations can be followed. Figures 3-6-7 can be made more understandable. Ethics committee support can be added to the method section.

Reply: Thanks for your comment. “Instructions for authors” of the journal “International Journal of Molecular Sciences” describes that the order of research manuscript sections is as follows: Introduction, Results, Discussion, Conclusions, and Materials and Methods in the section “Manuscript Preparation”. Figure 3, 6 and 7 were revised in the revised manuscript. The protocol No. was added in the statement of study approval in the revised manuscript.

In page 10 line 310 – 312 of the revised manuscript (changes are highlighted in red):

The experimental protocol (protocol No.: IACUC-18-221) was approved by the Institutional Animal Care and Use Committee of the National Defense Medical Center, Taipei, Taiwan. Animal care complied with institutional guidelines and regulations.

Reviewer 2 Report

Comments and Suggestions for Authors

The authors describe well designed experiments that indicate that in vitro C-PC reduces ROS production in H2O2 treated HEI-OC-1 cells and improves their viability. Furthermore, noise exposed guinea pigs were protected against noise induced damage to synapses between inner hair cells and spiral ganglion neurons and against oxidative stress in the spiral ligament.  There are a few concerns however, that need to be addressed by the authors.

METHODS

Why did the authors only test two concentrations of C-PC for their dose-response studies?

RESULTS

Since ROS levels were increased in the cochlea, why did the authors show increased expression of antioxidant enzymes in HEI-OC1 cells after H2O2 exposure?  Why were the expression of these enzymes not different in this cell line after concomitant treatment with C-PC compared to H2O2 alone (Figure 4) since they found that C-PC scavenges ROS in these cells?

p. 5 line 150. The authors stated: “restoration of IHC synaptic ribbons.”  I believe they should state: “preservation of IHC synaptic ribbons”.

FIGURE 8A AND 8B. From which turn of the cochlea were these mid-modiolar sections of the cochlea obtained?

Author Response

ijms-2979996 titled " C-Phycocyanin attenuates noise-induced cochlear synaptopathy via the inhibition of oxidative stress and intercellular adhesion molecule-1 in the cochlea "

Point-by-point responses to reviewer comments:

  1. The authors describe well designed experiments that indicate that in vitro C-PC reduces ROS production in H2O2 treated HEI-OC-1 cells and improves their viability. Furthermore, noise exposed guinea pigs were protected against noise induced damage to synapses between inner hair cells and spiral ganglion neurons and against oxidative stress in the spiral ligament.  There are a few concerns however, that need to be addressed by the authors.

METHODS

Why did the authors only test two concentrations of C-PC for their dose-response studies?

Reply: Thanks for your comment. It should be better to test more than two concentrations of the drug in a dose-response study. This study consists of experiments of the auditory cell line and animals. Due to the reduction of animals used in this research study (3Rs principle in the animal experiments), we determined only two concentrations of C-PC (1 and 5 μg/mL) tested in the experiments of the animals and cell culture.

  1. RESULTS

Since ROS levels were increased in the cochlea, why did the authors show increased expression of antioxidant enzymes in HEI-OC1 cells after H2O2 exposure?  Why were the expression of these enzymes not different in this cell line after concomitant treatment with C-PC compared to H2O2 alone (Figure 4) since they found that C-PC scavenges ROS in these cells?

Reply: Thanks for your comment. Different conditions of H2O2 treatment result in alterations of the activity of antioxidant enzymes [A-D]. The cells can respond to oxidants with an induction of their endogenous antioxidant enzymes [A]. In this study, H2O2 treatment leads to ROS generation and increased activity of antioxidant enzymes in HEI-OC1 cells. Increased activity of antioxidant enzymes in the cells results from the response of antioxidant defense system to ROS. C-PC can directly react to oxygen free radicals, such as alkoxyl, hydroxyl and peroxyl radicals and then scavenges ROS [10 in the revised manuscript]. In Figure 4, there was no significant difference in the activity of antioxidant enzymes between the groups treated with H2O2 alone and those treated with H2O2 and C-PC. Therefore, directly scavenging ROS is the mechanism of antioxidant activity of C-PC. We added the sentence “C-PC can react with oxygen free radicals and is an efficient scavenger of ROS [10]” to Results in the revised manuscript (page 3 line 107). The sentence “C-PC is a potent free radical scavenger and can react with alkoxyl, hydroxyl and peroxyl radicals” was added to Discussion in the revised manuscript (page 8 line 213).

In page 3 line 107 - 108 of the revised manuscript (changes are highlighted in red):

C-PC can react with oxygen free radicals and is an efficient scavenger of ROS [10].

In page 8 line 213 - 214 of the revised manuscript (changes are highlighted in red):

C-PC is a potent free radical scavenger and can react with hydroxyl and peroxyl radicals [10]

Reference:

[A] Röhrdanz E, Kahl R. Alterations of antioxidant enzyme expression in response

to hydrogen peroxide. Free Radic Biol Med. 1998 Jan 1;24(1):27-38.

[B] Wijeratne SS, Cuppett SL, Schlegel V. Hydrogen peroxide induced oxidative

stress damage and antioxidant enzyme response in Caco-2 human colon cells. J

Agric Food Chem. 2005 Nov 2;53(22):8768-74.

[C] Cho SI, Jo ER, Song H. Urolithin A attenuates auditory cell senescence by

activating mitophagy. Sci Rep. 2022 May 11;12(1):7704.

[D] Yu HH, Hur JM, Seo SJ, Moon HD, Kim HJ, Park RK, You YO. Protective effect of ursolic acid from Cornus officinalis on the hydrogen peroxide-induced damage of HEI-OC1 auditory cells. Am J Chin Med. 2009;37(4):735-46.

  1. p.5 line 150. The authors stated: “restorationof IHC synaptic ribbons.”  I believe they should state: “preservation of IHC synaptic ribbons”.

Reply: Thanks for your comment. “restoration of IHC synaptic ribbons.” was revised as “preservation of IHC synaptic ribbons” in the revised manuscript.

In page 5 line 150 - 151 of the revised manuscript (changes are highlighted in red):

The C-PC5 group exhibited more significant preservation of IHC synaptic ribbons than did the saline group.

  1. FIGURE 8A AND 8B. From which turn of the cochlea were these mid-modiolar sections of the cochlea obtained?

Reply: The representative figures are the basal turns of mid-modiolar sections of the cochlea.

In page 8 line 195 – 199 of the revised manuscript (changes are highlighted in red):

Figure 8. C-PC treatment attenuated noise-induced ROS generation and ICAM-1 expression in the cochlea. Representative immunohistochemical staining (brown color) for (A) 4-HNE and (B) ICAM-1 in the cochleae 48 h after noise exposure in the three groups. The basal turns of mid-modiolar sections of the cochleas are shown. The inset figure shows the region of the spiral ligament. n = 4 for each group. Scale bar: 100 μm.

Reviewer 3 Report

Comments and Suggestions for Authors

This study is well done and provides interesting new information. The only major concern is with their interpretation of results, as highlighted in the title. The authors do state in text that c-phycocyanin can have multiple influences including on oxidative stress and inflammatory response. They then show that it decreases the effects of noise on markers of oxidative stress and inflammatory response in cochlear tissues. However, the title and conclusions state it is the inhibition of oxidative stress that leads to decreased synaptopathy. They cannot rule out that is inhibition of inflammatory response that is responsible of even another influence of C-phy that they have not measured. While their results on HEI-OC1 support inhibition of oxidative stress for hair cell survival, this is not necessarily pertinent to synaptopathy. The authors need to be more cautious in their interpretations and discuss alternatives.

Minor concerns are on methodological details. They should provide rationale for choice of males only. How may subjects were used for each of the metrics done? Was there any blinding done (so that those doing metrics did not know the condition group of the subject they were assessing)?

Author Response

Point-by-point responses to reviewer comments:

  1. This study is well done and provides interesting new information. The only major concern is with their interpretation of results, as highlighted in the title. The authors do state in text that c-phycocyanin can have multiple influences including on oxidative stress and inflammatory response. They then show that it decreases the effects of noise on markers of oxidative stress and inflammatory response in cochlear tissues. However, the title and conclusions state it is the inhibition of oxidative stress that leads to decreased synaptopathy. They cannot rule out that is inhibition of inflammatory response that is responsible of even another influence of C-phy that they have not measured. While their results on HEI-OC1 support inhibition of oxidative stress for hair cell survival, this is not necessarily pertinent to synaptopathy. The authors need to be more cautious in their interpretations and discuss alternatives.

Reply: Thanks for your comment.

(1) The title and conclusion were revised in the revised manuscript. In addition, discussion about the anti-inflammatory effect of C-Phycocyanin was added as below.

In page 1 line 2 - 4 of the revised manuscript (changes are highlighted in red):

C-Phycocyanin Attenuates Noise-Induced Cochlear Synaptopathy via the Inhibition of Oxidative Stress and Intercellular Adhesion Molecule-1 in the Cochlea

In page 10 line 292 - 295 of the revised manuscript (changes are highlighted in red):

In our animal model of noise trauma, intratympanic administration of C-PC improved the recovery of IHC-SGN synapses from noise-induced cochlear synaptopathy by attenuating oxidative stress and ICAM-1 expression.

In page 9 line 247 - 257 of the revised manuscript (changes are highlighted in red):

These results revealed the protective effect of C-PC on IHC-SGN synapses. Moreover, C-PC administration diminished noise-induced oxidative stress and ICAM-1 expression in the cochlea. These findings demonstrated that C-PC can attenuate noise-induced cochlear synaptopathy by scavenging ROS and inhibiting ICAM-1. Excessive ROS formation is one main cause of noise-induced damage to IHC synaptic ribbons [33]. Therefore, the antioxidant activity of C-PC plays a crucial role in the protection of IHC-SGN synapses. ICAM-1 facilitates inflammatory response in the noise-exposed cochlea [26,27]. Since cochlear ICAM-1 expression can be inhibited by C-PC, C-PC exerts anti-inflammatory activity contributing to the protective effect of C-PC on the synapses. However, the effect of C-PC on inflammation-associated signaling pathways was not examined in this study and there is need for further investigation.

(2) The auditory cell line, HEI-OC1 cells, is a good model to test the protective/therapeutic effect of drugs on the cell damage [A]. Oxidative stress in the cochlea is a central mechanism leading to noise-induced hearing loss. ROS can damage IHC synaptic ribbons and have been demonstrated to be the cause of cochlear synaptopathy [33 in the manuscript]. Therefore, the purpose of experiments of HEI-OC1 cells is to investigate whether C-Phycocyanin can protect auditory cells against ROS-induced cell damage. According to the results of cell experiments, animal experiments were performed to investigate whether C-PC can attenuate oxidative stress in the cochlea and cochlear synaptopathy. The results of animal experiments have demonstrated the protective effect of C-Phycocyanin on noise-induced cochlear synaptopathy. Cochlear explant culture (an ex vivo model) is a better model to investigate the protective effect of C-Phycocyanin on the synapses before in vivo experiments are performed. However, sacrificing more animals is required for collecting cochlear explants. According to 3Rs principle in the animal experiments, we determined to perform experiments of HEI-OC1 cells instead of cochlear explant culture before the animal experiments.

Reference:

[A] Kalinec G, Thein P, Park C, Kalinec F. HEI-OC1 cells as a model for investigating drug cytotoxicity. Hear Res. 2016 May;335:105-117.

  1. Minor concerns are on methodological details. They should provide rationale for choice of males only. How may subjects were used for each of the metrics done? Was there any blinding done (so that those doing metrics did not know the condition group of the subject they were assessing)?

Reply: Thanks for your comment.

(1) In rodents, sex differences existed in age related-, noise induced hearing loss, ototoxicity, otoprotective therapy, sound therapy, auditory-related behaviors, and cochlear blood flow [A]. Sex differences should be considered in auditory experiments[A,B]. A previous study reports sex difference in ABR assessment of guinea pigs, with females exhibiting slightly higher ABR threshold for click and lower frequencies and lower ABR thresholds for higher frequencies than males [B]. Therefore, we used male guinea pigs (the animals with the same sex) in this study.

Reference:

  [A] Lin N, Urata S, Cook R, Makishima T. Sex differences in the auditory functions of rodents. Hear Res. 2022 Jun;419:108271.

  [B] Whitlon, DS, Young, H, Barna, M, Depreux, F, Richter, CP. Hearing differences in Hartley guinea pig stocks from two breeders. Hear Res. 2019:379:69-78.

(2) A total of 42 male guinea pigs were used in this study. One ear of each guinea pig was used in the experiments. We described the number of used animals in detail as below:

  • 24 guinea pigs for ABR test (n=8 in Figure 5) and subsequent confocal microscopic analysis of inner hair cells [8 of n(=10) in Figure 6 and 7]: 8 per group x 3 groups = 24
  • 6 guinea pigs for confocal microscopic analysis of inner hair cells [other 2 of n(=10) in Figure 6 and 7]: 2 per group x 3 groups = 6
  • 12 guinea pigs for immunohistochemical staining of the cochlea for 4-HNE and ICAM-1 (n=4 in Figure 8): 4 per group x 3 groups = 12

Total number: 24+ 6+ 12 = 42 animals

(3) Yes, the experiments were blindly performed. Those doing metrics did not know the condition group of the subject they were assessing.

Round 2

Reviewer 1 Report

Comments and Suggestions for Authors

Thanks to author's effort and revision. the manuscript is acceptable as this form